# Difunctional Magnetic Nanoparticles Employed in Immunochromatographic Assay for Rapid and Quantitative Detection of Carcinoembryonic Antigen

**DOI:** 10.3390/ijms241612562

**Published:** 2023-08-08

**Authors:** Yalin Hu, Xin Lu, Liyue Shen, Jiahui Dong, Zhanwei Liang, Jie Xie, Tao Peng, Xiaoping Yu, Xinhua Dai

**Affiliations:** 1College of Life Sciences, China Jiliang University, Hangzhou 310018, China; s21090710017@cjlu.edu.cn (Y.H.); p22091055042@cjlu.edu.cn (L.S.); s22090710017@cjlu.edu.cn (J.D.); yxp@cjlu.edu.cn (X.Y.); 2Technology Innovation Center of Mass Spectrometry for State Market Regulation, Center for Advanced Measurement Science, National Institute of Metrology, Beijing 100029, China; 13331218159@163.com (X.L.); s20090710033@cjlu.edu.cn (Z.L.); xiejie@nim.ac.cn (J.X.)

**Keywords:** magnetic nanoparticles, immunochromatographic detection, carcinoembryonic antigen

## Abstract

Immunochromatographic assay (ICA) plays an important role in in vitro diagnostics because of its simpleness, convenience, fastness, sensitivity, accuracy, and low cost. The employment of magnetic nanoparticles (MNPs), possessing both excellent optical properties and magnetic separation functions, can effectively promote the performances of ICA. In this study, an ICA based on MNPs (MNP–ICA) has been successfully developed for the sensitive detection of carcinoembryonic antigen (CEA). The magnetic probes were prepared by covalently conjugating carboxylated MNPs with the specific monoclonal antibody against CEA, which were not only employed to enrich and extract CEA from serum samples under an external magnetic field but also used as a signal output with its inherent optical property. Under the optimal parameters, the limit of detection (LOD) for qualitative detection with naked eyes was 1.0 ng/mL, and the quantitative detection could be realized with the help of a portable optical reader, indicating that the ratio of optical signal intensity correlated well with CEA concentration ranging from 1.0 ng/mL to 64.0 ng/mL (R^2^ = 0.9997). Additionally, method comparison demonstrated that the magnetic probes were beneficial for sensitivity improvement due to the matrix effect reduction after magnetic separation, and the MNP–ICA is eight times higher sensitive than ICA based on colloidal gold nanoparticles. The developed MNP–ICA will provide sensitive, convenient, and efficient technical support for biomarkers rapid screening in cancer diagnosis and prognosis.

## 1. Introduction

Point-of-care testing (POCT) is an economical, rapid, and reliable medical in vitro diagnostic technology that provides results in a short time to save detection time under limited resources with the help of portable diagnostic tools. It has gradually penetrated people’s lives for self-health monitoring and disease diagnosis in many developing countries and remote areas [1,2]. According to the classification of detection platforms, POCT techniques include immunochromatographic assays (ICAs) [3,4], microfluidic chips [5,6], electrochemical biosensors [7,8], and photoelectrochemical biosensors [9]. Among them, ICA based on colloidal gold nanoparticles (CGNP-ICA) has become one of the most well-known POCT techniques amidst the wave of SARS-CoV-2 sweeping the world because it has the advantages of low cost, simple operation, strong specificity, and fast detection speed, which effectively alleviated the detection pressure of large amount samples [10]. CGNPs are considered reliable label materials in ICA tests because of their easy preparation, adjustable size, and biocompatibility. Additionally, gold nanoclusters, gold nanorods or gold nanoparticles coupled metal particles [11], metal oxides [12], metal–organic frameworks [13], quantum dots [14], and magnetic nanoparticles (MNPs) have been applied to the ICAs for performances improving, and great progress has been achieved.

Among them, MNPs have attracted extensive attention in the fields of medicine and biology, such as in vitro diagnosis, drug tracking and targeted administration, targeted therapy, and magnetic resonance imaging [15,16]. To develop a sensitive and accurate detection method based on ICA, MNPs with stable physical properties have been employed as labels. Compared with the purification of the traditional colloidal gold probes with long-term centrifugation, magnetic probes could be purified simply and rapidly by an external magnetic field within several minutes, and they could be quickly redispersed when the magnetic field is removed. Thus, the magnetic probes are not only employed to enrich and extract targets from samples but also used as a signal output with its inherent optical property (bright brown color). In recent years, many MNP-based ICA methods have been widely developed for target detection based on their optical, magnetic, and photothermal signals [17]. Chen et al. successfully synthesized a “three-in-one” multifunctional magnetic nanocomposites composed of MNPs, MIL-100(Fe), and platinum nanoparticles, which were used as the labels for ICA method development, and the sensitivity of the developed novel ICA for procalcitonin detection was about 2280 times higher than that of the traditional GCNP–LFIA [18]. Wang et al. described a sensitive and quantitative ICA strip based on magnetic fluorescent nanoparticles for bacterial detection; the proposed method exhibited high sensitivity for *Streptococcus pneumoniae* detection, suggesting it was nearly 1000 times more sensitive than ICA based on quantum dots [19]. In general, the magnetic separation function of MNPs contributes greatly to sensitivity improvement; MNPs–ICA may be a promising tool for tumor biomarker testing.

ICA methods in POCT can be used as supplementary means of medical diagnosis to relieve pressure on the health care system and patients. This is of great significance for assisting in early screening and diagnosis, efficacy prediction, and prognosis [20]. Carcinoembryonic antigen (CEA), as a broad-spectrum tumor biomarker, is a reliable indicator of lung cancer [21], gastric cancer [22], colorectal cancer [23], pancreatic cancer [24], and breast cancer [25]. Generally, an abnormal health condition will be indicated when the average CEA concentration is more than 5 ng/mL [26], and it was reported that the CEA levels in cancer patients’ serum were significantly higher than those in non-cancer patients’ serum [27]. Thus, sensitive and accurate detection of CEA is of great value to clinical diagnosis. At present, electrochemical biosensors [28,29], piezoelectric biosensors [30], 18F-fluorodeoxyglucose emission tomography/computed tomography (FDG-PET/CT) [31], enzyme-linked immunosorbent assay [32], and ICA methods have been developed for CEA detection. Except for the ICA method, most methods are expensive, time-consuming, and complex to operate. In this work, integrating with the dual-antibody sandwich model, an ICA method based on MNPs has been proposed to sensitively detect CEA in serum. As illustrated in Figure 1, MNPs modified with abundant carboxy groups (MNP–COOH) were coupled with monoclonal antibody I against CEA (mAbI) by the active ester method (Figure 1A). Then, the prepared magnetic probes (MNP–mAbI) were mixed and incubated with the samples to capture CEA, resulting in the formation of “MNP-mAbI-CEA” immunocomplexes, which were collected and washed under the help of an external magnetic field (Figure 1B). Finally, the immunocomplexes were redispersed with buffer and added to the sample pad of the MNP–ICA strip for CEA detection (Figure 1C). Theoretically, distinct brown bands appear on the test and control lines when CEA is present in the serum sample, and conversely, there is only a brown band on the control line (Figure 1D). Therefore, the brown color intensity on the test line is a positive correlation with the concentration of CEA in the serum sample. Simultaneously, the color intensity can be recorded by a portable reader for CEA quantitative detection. Results demonstrate that the developed MNP–ICA exhibits simplicity, convenience, high sensitivity, and selectivity for CEA detection and is a promising tool for other tumor biomarkers screening.

## 2. Results and Discussion

### 2.1. Characterization and Validation of MNP–mAbI Preparation

Theoretically, the MNPs modified with abundant carboxyl groups, exhibiting good dispersion with an average diameter at about 300 nm (Figure 1A), were covalently conjugated with mAbI via the active ester method. According to Section 2.3, during the process of MNP–mAbI preparation, other protein or stabilizers have not been added in order to eliminate the conjugation of other proteins on the MNPs for characterization. Figure 1B displays that the hydrodynamic diameter of MNPs increased from 448 nm to 596 nm after conjugating with mAbI, and the zeta potential in ultrapure water decreased from −15.391 mV to −19.268 mV (Figure 1C), higher electronegativity makes MNP–mAbI disperse well in aqueous solutions and be not prone to agglomerate. In addition, a vibrating sample magnetometer was applied to measure the saturation magnetization, as shown in Figure 1D, the saturation magnetization of MNPs was 58.9 emu/g while it decreased to 34.1 emu/g after mAbI attached, which indicates that the magnetic shielding effect has been introduced because of the conjugation between MNPs and mAbI, but it has no significant effect on the magnetic separation performance under the external magnetic field. These results demonstrate that mAbI has been conjugated onto the surface of MNPs, and the magnetic detection probes have been prepared successfully.

### 2.2. Parameters Optimization for MNP–ICA Development

In order to obtain better performances of the MNP–ICA, the parameters have been optimized by using a single variable method. Results are presented in Appendix A; the optimal parameters for MNP–mAbI preparation were as below, (1) the pH was 7.5, (2) both EDC and NHS dosages were 150 μg, (3) the mAbI amount for labeling was 10 μg. And the concentrations of mAbII and goat anti-mouse IgG on the T and C lines of the MNP–ICA strip was 1.0 mg/mL and 0.2 mg/mL, 3.0 μL of MNP–mAbI was required for each MNP–ICA strip.

Because MNPs exhibit both excellent optical properties (bright brown) and unique magnetic separation performance, the prepared MNP–mAbI can be employed to enrich CEA in the serum samples and reduce the influence of the matrix effect for the MNP–ICA detection. As shown in Appendix A, when different concentrations of CEA (5.0 ng/mL, 25 ng/mL, 50 ng/mL, 75 ng/mL, and 100 ng/mL) were detected, compared to the MNP–mAbI applied directly without incubation, stronger signal intensities on the T and C lines of MNP–ICA strip were generated by employing MNP–mAbI to incubate with samples before detection, which was validated by the obtained and recorded signal intensities of T lines. It demonstrated that the preincubation between MNP–mAbI and samples was beneficial in improving the sensitivity of the MNP–ICA. Sequentially, the incubation time also has been optimized, and 5 min was selected (Appendix A). In addition, the immunoreaction on the MNP–ICA test strip is relative to the signal intensity, detection sensitivity, and detection efficiency. The brown color on the T and C lines deepened with the time increasing, but the ratio of T and C line signal intensity trended to balance and remained basically the same from 10 min to 25 min (Appendix A), 10 min was optimized as the immunoreaction time on the ICA strip. Thus, the detection of CEA in serum with the developed MNP–ICA can be completed within 15 min.

### 2.3. Performances of the MNP–ICA for CEA Detection

CEA standard solution was diluted with human serum to a series concentration (0.0, 1.0, 2.0, 4.0, 8.0, 16.0, 32.0, 64.0, 128.0, and 256.0 ng/mL) and detected by the developed MNP–ICA under the optimal parameters. The brown color on the T lines deepened, but that on the C lines weakened as the concentration increased because more and more MNP–mAbI were bound with CEA and captured by the mAbII coated on the T line. And the brown band on the T line can be clearly distinguished by the naked eye even though the CEA concentration was as low as 1.0 ng/mL (Figure 2A). The colorimetric signal intensities on the T and C lines (OD_T_ and OD_C_) were obtained and recorded by a portable reader, of which alteration tendencies were consistent with the results observed by the naked eye. It is reported that the OD_T_/OD_C_ ratio can effectively offset the effects of the inherent heterogeneity of test strips and the matrix containing the samples [33]. The OD_T_/OD_C_ ratios were calculated and exhibited better precision than OD_T_ (Appendix A). A sigmoidal curve fitted by plotting the OD_T_/OD_C_ ratio against the CEA concentration is displayed in Figure 2B, which indicates the OD_T_/OD_C_ ratio correlated well with logarithmic concentration from 1.0 ng/mL to 64.0 ng/mL, the regression equation is y = 1.066 − 0.956/(1 + (x/9.390)^1.056^) (R^2^ = 0.9997). The limit of detection (LOD) of MNP–ICA was calculated according to the regression equation and formula: *I_min_* = $\bar{X}$ + 3 × σ, where *I_min_* is the OD_T_/OD_C_ ratio corresponding to the LOD, $\bar{X}$ is the average OD_T_/OD_C_ ratio of 11 blank serum samples, σ is the standard deviation [34]. As calculated, the LOD was 0.53 ng/mL, which is about twice lower than that of the assay mode without incubation (Appendix A).

Several common tumor biomarkers in serum, such as AFP, Cyfra21-1, GDF-15, NSE, CEA, CA125, and CA153, have been detected by the MNP–ICA to investigate the specificity. As displayed in Appendix A, the T lines basically have no colorimetric signal similar to the blank sample, except that an obvious brown band appeared on the T line when 10 ng/mL of CEA was detected, which suggested that the developed MNP–ICA is highly selective and sensitive to CEA in serum.

Blank serum samples spiked with CEA at high (64.0 ng/mL), medium (16.0 ng/mL), and low (1.0 ng/mL) concentrations have been tested three times a day for three days, and the relative standard deviation (RSD) was calculated to evaluate the intra-assay and inter-assay variations of the MNP–ICA. As presented in Table 1, RSD for intra-assay was less than 14.8%, and that for inter-assay ranged from 9.6% to 19.7%. Additionally, the recoveries between 78.1% and 104% indicated the MNP–ICA is accurate for CEA detection.

### 2.4. Method Comparisons and Human Serum Samples Detection

GCNPs are the common and stable labels in the ICA method; an ICA method base on homemade GCNPs (characterized in Appendix A) has been developed with the same antibody pair and employed to detect the series of CEA dilutions. Figure 3A shows that the red band on the T line of GCNP–ICA appeared when the CEA concentration was 8.0 ng/mL, indicating the qualitative sensitivity of GCNP–ICA is eight times lower than that of MNP–ICA. Sequentially, the signal intensities of T lines on GCNP–ICA and MNP–ICA were compared, as displayed in Figure 3B, indicating that the optical signal of MNPs is stronger than that of GCNPs, which could explain the sensitivity of MNP–ICA is superior to GCNP–ICA. As for the quantitative detection of GCNP–ICA, a good linear range for the quantitative detection was 8.0–512.0 ng/mL (Appendix A), and the LOD was evaluated and calculated as 7.41 ng/mL, which was also much higher than that of MNP–ICA. Additionally, some biosensors and ICA methods based on nanomaterials reported previously [35,36,37,38,39,40] have been summarized in Appendix A, suggesting that the performances of the developed MNP–ICA are equal or superior to them.

Serum samples from five healthy people and five liver cancer patients were tested by the MNP–ICA. The results shown in Figure 3C indicate that the color signals on T lines for liver cancer patients’ detection differed significantly from healthy ones. The signals were converted into CEA concentration according to the regression equation (y = 1.066 − 0.956/(1 + (x/9.390)^1.056^); as plotted in Figure 3D, the CEA levels in healthy people are lower than or close to the LOD, and those in patients are much higher than 1.0 ng/mL, which indicated that the developed MNP–ICA is adequate for rapidly and quantitatively detection of CEA level in human serum.

## 3. Experimental

### 3.1. Chemicals and Materials

CEA antigen was purchased from Novoprotein Scientific Co., Ltd. (Suzhou, China). Monoclonal antibodies I and II against CEA (mAbI and mAbII) were purchased from Hefei Kangruixiang Biotechnology Co., Ltd. (Hefei, China). Goat Anti-Mouse IgG was purchased from Jackson ImmunoResearch Laboratories, Inc. (West Grove, PA, USA). Carboxylated MNPs (300 nm, coated polystyrene), N-(3-Dimethylaminopropyl)-N’-ethylcarbodiimide hydrochloride (EDC), N-Hydroxysuccinimide (NHS), Polyethylene glycol, Tween-20, and Potassium carbonate were acquired from Aladdin Reagent Co., Ltd. (Shanghai, China). 2-(N-morpholino) ethanesulfonic acid (MES) was supplied by Vdo Biotechnology Co., Ltd. (Suzhou, China). Borate buffer solution was purchased from Coolaber Science & Technology Co., Ltd. (Beijing, China). Sodium chloride was purchased from Sinopharm Chemical Reagent Co., Ltd. (Shanghai, China). Bovine serum albumin (BSA), ProCline-300, Tetronic 1307, Sucrose, D-(+)-Trehalose dihydrate, and Polyvinylpyrrolidone were obtained from Sigma-Aldrich Chemical Corporation (St. Louis, Mo, USA). Phosphate Buffer Saline Tablets and Tris-HCl buffer were purchased from Solarbio Science & Technology Co., Ltd. (Beijing, China). GCNPs (~40 nm) solution was prepared in our laboratory [41]. Polyvinyl chloride (PVC) pad was acquired from Bituo New Materials Co., Ltd. (Qingdao, China). The absorbent pad, sample pad, and binding pad were purchased from Kinbio Tech Co., Ltd. (Shanghai, China). Nitrocellulose (NC) membrane was purchased from Sartorius (Göttingen, Germany).

### 3.2. Apparatus and Characterization

The microcomputer automatic cutting machine (ZQ2402), CNC cutting machine (CTS300), and XYZ 3D film spraying instrument (HM3035) were supplied by Kinbio Tech Co., Ltd. (Shanghai, China). Ultrapure water was purified with Milli-Q system from Millipore Corp. (Bedford, MA, USA). Electronic balance was purchased from Mettler Toledo Instruments Co., Ltd. (Shanghai, China). Portable reader was supplied by HYK Gene Technology Co., Ltd. (Shenzhen, China); the principle of portable reader for signal readout is as follows: a high-performance LED with a wavelength of 500–520 nm was used as light source, and the light was focused on the MNP–ICA test strip by optical lens. According to reflectance spectroscopy, the MNPs on the test strip would produce optical reflection signal due to the strong absorbance of MNPs. The produced reflective optical signal was converted into photoelectrical signals by conversion module.

### 3.3. Preparation of Magnetic Detection Probes

The magnetic detection probes (MNP–mAbI) were prepared by MNPs covalently conjugating with mAbI. Briefly, 0.5 mg of MNPs were dissolved in 250 μL of MEST buffer (MES buffer containing 0.05% Tween-20) after washing three times. Moreover, 50 μL of NHS (3 mg/mL) and EDC (3 mg/mL) were added in succession to activate the carboxyl groups on the surface of MNPs. Stirring continuously for 30 min at room temperature, the MNPs were washed three times with 5 mM BBT buffer (borate buffer containing 0.05% Tween-20, pH7.5) and then resuspended with 250 μL of BBT buffer. Then, 100 μL of mAbI (0.1 mg/mL) was added and agitated at room temperature for 1 h. BBT buffer was removed with magnetic separation, 250 μL of 1% PEG solution containing 0.05% Tween-20, and 250 μL of 10% BSA solution containing 0.05% Tween-20 were added to block the nonspecific sites. After 30 min, with the help of external magnetic field, the supernatant was discarded, and the precipitate was redispersed in 100 μL of 0.02 M Tris-HCl buffer (containing 0.5% trehalose, 10% sucrose, 0.5% PVP, 0.1% S9, 0.05% Proclin-300, 1% BSA, 0.1% Tween-20), stored at 4 °C until use.

### 3.4. Fabrication of the MNP–ICA Test Strip

The mAbII and goat anti-mouse IgG were diluted with PBS buffer (0.01 M, pH 7.4), sprayed on the NC membrane with 0.8 μL/cm as the test (T) and control (C) lines, and then dried at 37 °C for 12–16 h. The structure of MNP–ICA test strip is illustrated in Figure 1C. The NC membrane, conjugated pad, sample pad, and absorption pad were successively pasted on the PVC backing pad by overlapping 2.0 mm. The fabricated MNP–ICA test strip was cut into 3.0 mm wide strips, stored at room temperature, and kept dry.

### 3.5. Processes of the MNP–ICA for CEA Detection

As illustrated in Figure 1B, 3 μL of MNP–mAbI was mixed with 10 μL of serum sample and incubated for 5 min, the serum was removed, and the formed magnetic immunocomplex was washed three times with the help of external magnetic field. The obtained collection was dispersed with 80 μL of PBS buffer (containing 3% NaCl, 1% Tween-20, and 1% BSA) and then dropped onto the sample pad of MNP–ICA strip. After 10 min, the qualitative results can be observed by naked eye, and the quantitative detection results can be obtained with the help of the portable reader.

The processes of GCNP–ICA were described in the Appendix A.

## 4. Conclusions

In this work, a rapid and sensitive ICA method based on MNPs has been successfully established for CEA detection, in which the prepared magnetic probes MNP–mAbI exhibited excellent magnetic responsiveness and optical property and were employed for target enrichment and signal output, allowing the qualitative and quantitative detection of CEA in serum to be finished within 15 min. The sensitivity of the MNP–ICA was eight times higher than that of the GCNP–ICA developed with the same antibody pair in our work, attributing to (1) the magnetic probes can effectively reduce the serum matrix effect by magnetic separation and (2) the optical signal intensity of MNPs is stronger than that of GCNPs. The results demonstrated that the developed MNP–ICA is convenient, high-efficiency, sensitive, specific, and accurate for CEA in serum determination, and it is also suitable for screening other disease markers, which has potential applications for social and economic benefits.

## Data Availability

The data presented in this study are available on request from the corresponding author. The data are not publicly available due to privacy restrictions.

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
