# Peer review of "Difunctional Magnetic Nanoparticles Employed in Immunochromatographic Assay for Rapid and Quantitative Detection of Carcinoembryonic Antigen"

_ijms, 2023, doi:10.3390/ijms241612562_

Round 1
Reviewer 1 Report
Comments and Suggestions for Authors
Yalin Hu and colleagues presented an immunochromatographic test for carcinoembryonic antigen in human serum, which utilizes magnetic nanoparticles for sample enrichment and color generation. The results of the assay can be evaluated visually or using a portable reader. I find this paper to be intriguing, with well-presented and clear results. However, I believe that certain statements and conclusions require further supporting evidence. Please see my specific comments below.
1. Line 53-54 – Please, cite papers on instability of gold labels in LFIA. As far as I know, gold conjugates are usually stored in dried state inside conjugate pad of the strip and can be stored for a long time. Without supporting references argument 1 should be modified or removed.
2. Line 54 – argument 2. Indeed, certain labels, particularly catalytic ones, can significantly decrease the limit of detection (LOD) compared to gold nanoparticles. However, considering the broad readership of IJMS, it is possible that readers may be unfamiliar with this fact. Therefore, please find a reference to support this statement.
3. Line 55 – “the colorimetric signal of GCNPs are insufficient” in argument 2 the Authors have already mentioned insufficient LOD in gold-based LFIA. What is the difference between arguments 2 and 3? Moreover, they referenced a paper (Ref 13) on the comparison of gold-based lateral flow immunoassay (LFIA) and nanozyme-based LFIA, in which gold nanoparticles without amplification provided a limit of detection (LOD) comparable to that generated by the nanozyme with amplification. I am not sure if this reference is relevant in this context. Line 56 – “the colorimetric signal … easily affected by the pH of the system [14]” Ref. 14 does not contain any information about issues with pH in gold-based LFIA. Remove this paper and cite relevant one. In total, both references supporting argument 3 are irrelevant. Therefore, argument 3 should be reconsidered or removed.
4. Line 58 – “organic frame metal materials” I think “metal-organic framework” should be used here
5. Line 91 - check «detection,»
6. Line 149 – what is S9?
7. Line 170 – conjugated nanoparticles are very large, almost 600 nm. Please, provide values of polydispersity index and, if possible, their microscopy (SEM, TEM, or AFM) images.
8. Fig. 1 – please specify buffer in which zeta potential of nanoparticles has been measured.
9. Line 177-179 – strictly speaking, change of size, zeta potential and magnetic properties do not confirm attachment of antibodies. All these changes can be result of chemical reaction with ester and adsorption of BSA and other stabilizers.
10. Figure S1 – what is “+” and “-” sample?
11. Line 195 – how can you explain increase of C line intensity at longer incubation times? Control line is anti-mouse antibodies. How can incubation of MNP-Mab in human serum affect their interaction with goat antibodies? It looks like non-specific interactions.
12. Line 195 – given that the authors used as a signal output ratio of T/C intensities rather than intensity of T line alone, incubation in serum, which increases BOTH intensity of T and C lines seems non-profitable.
13. Line 197-198 – the Authors demonstrated that incubation increased intensity of T line (Fig. S2C), but they did not show better sensitivity (LOD). In negative sample, I clearly see signal in T line for both assay modes (with and without incubation). LOD depends on both signal intensity of test sample and signal in blank (negative sample). Authors should calculate LODs for both assay modes (with and without incubation).
14. Inset of Fig. 2 – I recommend the Authors fit their data to sigmoidal curve (4-PL logistic) for better accuracy at lower concentrations (see table 1).
15. Please, describe assay with gold nanoparticles in Methods section. Was the volume and dilution of serum sample the same for both tests?
16. Did you optimized GCNP-ICA as thoroughly as MNP-ICA? 8-fold decrease of visual LOD can be explained by suboptimal assay conditions. There are numerous parameters that affect GCNP-ICA results: size of gold nanoparticles, amount of antibodies, conjugation conditions, amount of capture antibodies and so on. Were these parameters optimized?
Reviewer 2 Report
Comments and Suggestions for Authors
The manuscript describes the design of a lateral flow assay labelled with MNPs-Ab conjugates. The tracer also acts as a magnetic nanosorbent for pre-analytical analyte (CEA) separation from complex sample. The manuscript is well written, although significant revisions are required for it to be considered for publication in IJMS journal.
1) As there are a number of similar LFA solutions based on MNPs as labels, it is advisable to clearly highlight the elements of novelty and emphasise the originality of the presented research.
2) Lines 53-59: I disagree with the Authors who are discrediting GCNPs as difficult to conjugate and contrasting them with MNPs as easy to conjugate. There are many stable (even commercial) conjugates employing covalent coupling to the gold surface via thiol chemisorption (see. https://nanocomposix.com/collections/material-gold/shape-spheres) MNPs themselves are not compatible with carbodiimide chemistry - their reactivity (as with other nanomaterials) is determined by the surface coating e.g. here gold is used as a compatibiliser to facilitate the functionalisation of MNPs: 10.1016/j.bios.2023.115511).
3) The introduction section could have been a somewhat shorter and more focused on immunochromatographic assays and CEA rather than such a broad yet superficial review of a number of aspects loosely related to the essence of the manuscript.
4) It is advisable to state relatively early and clearly in the text what is the diagnostically relevant range of CEA concentrations.
5) The Experimental section lacks some crucial information that should be completed:
- more information on MNPs and GCNPs is required. What was the manufacturer's declared diameter and surface coating of MNPs?
- Information is missing regarding colloidal gold nanoparticles (manufacturer, size, surface coating etc.) as well as the protocol for their modification. Was this protocol as well as the whole immunochromatographic assay also optimised for GCNPs similarly as for MNPs? Because maybe the difference in LOD is due to suboptimal conditions?
6) Lines 145-146: there is no information on the volume of the BSA with Tween-20 solution added.
7) Line 149: what is 0.1% of S9?
8) I consider that making conclusions about effective modification based on changes in hydrodynamic diameter and zeta potential is not justified. Have the size + zeta analyses been repeated? If not, they should have, as MNPs samples are difficult to be analysed due to limited stability.
9) The principle of the readout with the portable reader needs to be clarified. Does the readout is from a selected area or from the whole width of the strip (the intensities of colour may be on-uniform)?
10) Why did the Authors use a normalised signal (ODT/ODC) so consistently? Did calibration curves based directly on ODT values have significantly inferior performance? Please provide a more extensive explanation in the manuscript.
11) GCNPs and MNPs are rather poorly characterised. Therefore, it is difficult to clearly assess what is the source of the "excellent optical property" of magnetic nanoparticles suggested by the Authors. Are values of molar extinction coefficients of both nanolabels known?
12) Fig. 3D: What is the repeatability of the results shown in Fig. 3B? A single measurement can easily distinguish between CEA levels in patients and healthy individuals, but an assessment of repeatability (providing error bars) would allow a better evaluation of the potential of the method in quantitative analysis of CEA levels.
Round 2
Reviewer 1 Report
Comments and Suggestions for Authors
The authors did a great job and significantly improved their manuscript. I have only a few minor comments.
Line 179 - In Section 2.3. addition of various stabilizers is described, therefore the statement "According to the Section 2.3, MNP-mAbI was prepared without Formatted: Font: Italic addition of other protein or stabilizers in the process." is confusing.
Figure 1 - Please decipher DPI in figure caption
Comments on the Quality of English LanguageLine 178 - Wit
Fig. 1. "Characterizations of scanning electron microscope (A)" Characterization of microscope? Maybe "SEM image" of MNP?
Reviewer 2 Report
Comments and Suggestions for Authors
Dear Authors. Thank you for your prepared responses and revisions. After careful consideration, I still have a few comments and reservations, which I have indicated below:
Point 5: which method was used to characterize the GCNPs? (I found no confirmation in this or the Authors' original manuscript). 40 nm is a large diameter, bearing in mind that the synthesis method is a modified Turkevich-Frens approach. Besides, it is difficult to compare nanoparticles so different in many respects. including size, method of conjugation (chemical vs. passive adsorption), the concentration of different types of nanoparticles is also unknown and difficult to control....
Please provide experimental confirmation of the size of the nanoparticles and information on the concentration of the gold nanoparticles (at least information that the gold nanoparticles were not diluted, preferably analysis and calculations based on UV-Vis absorption spectra - see 10.1021/ac0702084).
Point 5: Thank you for the explanation supported by the literature reference. However, I am curious if the authors even tried to construct calibration curves on the basis of T-line signals only? And if they did and their performance was worse, an appropriate brief comment in the manuscript would be valuable.
Point 11: Since the optical properties of the nanoparticles are not known, the more I suggest recording the absorption spectra for both types of nanoparticles (they can be placed in Supplementary materials) and giving the concentrations for which the spectra of magnetic and gold nanoparticles were recorded. This will be the basis for comparing their optical properties.
Round 3
Reviewer 2 Report
Comments and Suggestions for Authors
Thank you for the work you put into revising the manuscript. After revisions, I can recommend it for publication in IJMS journal.